# Human Genetic Research in Wallacea and Sahul: Recent Findings and Future Prospects

**DOI:** 10.3390/genes13122373

**Published:** 2022-12-16

**Authors:** Leonard Taufik, João C. Teixeira, Bastien Llamas, Herawati Sudoyo, Raymond Tobler, Gludhug A. Purnomo

**Affiliations:** 1Australian Centre for Ancient DNA, School of Biological Sciences, University of Adelaide, Adelaide, SA 5005, Australia; 2Centre of Excellence for Australian Biodiversity and Heritage, University of Adelaide, Adelaide, SA 5005, Australia; 3Mochtar Riady Institute for Nanotechnology, Tangerang 15810, Indonesia; 4Evolution of Cultural Diversity Initiative, Australian National University, Canberra, ACT 2601, Australia; 5Centre for Interdisciplinary Studies, University of Coimbra, 3004-531 Coimbra, Portugal; 6Environment Institute, University of Adelaide, Adelaide, SA 5005, Australia; 7National Centre for Indigenous Genomics, Australian National University, Canberra, ACT 2601, Australia; 8Indigenous Genomics Research Group, Telethon Kids Institute, Adelaide, SA 5001, Australia

**Keywords:** Wallacea, Sahul, human migrations, phylogeography, ancient DNA

## Abstract

Genomic sequence data from worldwide human populations have provided a range of novel insights into our shared ancestry and the historical migrations that have shaped our global genetic diversity. However, a comprehensive understanding of these fundamental questions has been impeded by the lack of inclusion of many Indigenous populations in genomic surveys, including those from the Wallacean archipelago (which comprises islands of present-day Indonesia located east and west of Wallace’s and Lydekker’s Lines, respectively) and the former continent of Sahul (which once combined New Guinea and Australia during lower sea levels in the Pleistocene). Notably, these regions have been important areas of human evolution throughout the Late Pleistocene, as documented by diverse fossil and archaeological records which attest to the regional presence of multiple hominin species prior to the arrival of anatomically modern human (AMH) migrants. In this review, we collate and discuss key findings from the past decade of population genetic and phylogeographic literature focussed on the hominin history in Wallacea and Sahul. Specifically, we examine the evidence for the timing and direction of the ancient AMH migratory movements and subsequent hominin mixing events, emphasising several novel but consistent results that have important implications for addressing these questions. Finally, we suggest potentially lucrative directions for future genetic research in this key region of human evolution.

## 1. Introduction

The Wallacean archipelago and the paleocontinent of Sahul mark the farthest south-eastern destination for historical human migrant groups dispersing from Africa. Despite this remote location, local evidence from archaeological sites and the fossil record point to the presence of multiple distinct human groups in the region from at least 1.5 million years ago (Mya), including the oldest known hominin remains outside of Eurasia; i.e., *Homo erectus* in the island of Java [1,2]. While *H. erectus* likely reached Java while it was still contiguous with mainland Asia due to lower sea levels during the mid-Pleistocene, further east lies the maritime barrier known as Wallace’s Line, which has separated mainland Asia from the Wallacean and Philippine archipelagos and Sahul throughout hominin evolutionary history. This persistent barrier was responsible for the diversification of marsupial and placental mammals around 50 Mya, though notably, it did not prevent multiple hominin groups from venturing further east (Figure 1). Indeed, by around 700 thousand years ago (kya), a small-bodied and now extinct hominin known as *H. floresiensis* had moved into the Indonesian island of Flores [3,4,5], while *H. luzonensis* inhabited Lúzon, the northern island of the Philippine archipelago [6]. The three now-extinct human lineages conceivably survived in the region until the arrival of anatomically modern humans (AMH) migrating from Africa, with dates from AMH-associated sites in west Indonesia (Sumatra) [7] and northern Australia suggesting that initial settlement may have been as early as 70–65 kya [8,9,10,11,12].

Crucially, efforts to obtain robust estimates of the timing and direction of the historical migratory movements that eventually brought AMH and other hominin groups into Sahul have been complicated by the sparse archaeological record in Wallacea. Indeed, the current oldest evidence for AMH occupation in Wallacea recorded in Timor [11,13] and Sulawesi [14] at ~44 kya substantially postdates the aforementioned earliest archaeological evidence for AMH arrival in the region [9,10,15]. Further, much of our current understanding of the initial settlement patterns of Wallacea and Sahul has come from models that have used island intervisibility and paleogeographic reconstructions to infer optimal migration paths. These studies provide support for two alternate routes that take either a northern or southern route across Wallacea ([12,16,17,18]; also see Figure 1 for more details), suggesting that multiple historical paths were feasible.

With the current limited availability of archaeological sites across much of Wallacea, researchers have recently turned to genetic sequence data from local modern populations and ancient remains as an alternative empirical tool to help resolve fundamental questions around AMH arrival and hominin mixing events in the region. In the following sections, we review key findings from the small but growing number of population genetic and phylogeographic studies examining the human history in Wallacea and Sahul, with a particular focus on studies of Wallacean populations from the past decade. These studies have begun to illuminate a complex population history marked by multiple migrations and population mixing events, which may have erased much of the ancestry descending from the original AMH founders. We discuss the implications of these findings for our understanding of the broader human and hominin regional history and conclude by highlighting potentially fruitful areas of future research.

## 2. Genetic Evidence for AMH Arrival in Wallacea and Sahul

While human genetic research has accumulated steadily for populations in most parts of the world, comparatively few population genomic and phylogeographic studies leveraging complete mitogenome (i.e., mtDNA) sequences or Y-chromosomes currently exist for present-day Indigenous peoples from the former continent of Sahul [20,21,22,23,24,25,26,27]. Genomic data have repeatedly demonstrated that all contemporary non-African AMH populations have diversified from an ancestral AMH group that left Africa between 60–50 kya [28]; however, the initial results from a single deeply sequenced Aboriginal Australian genome derived from a ~100-year-old hair sample proposed that Indigenous Australians also carry substantial AMH ancestry from an earlier African diaspora that originated 75–62 kya [29]. While a similarly deep AMH ancestry source was reported for modern Papuans [30,31], other studies suggested that these may be an artefact stemming from modern Australo-Papuans who have up to 6% additional ancestry from one or more hominin groups related to Denisovans (an extinct hominin group first discovered through ancient DNA analysis from a finger bone fragment found in Siberia, also see the penultimate section below; [32,33,34]). Population genetic models that account for this Denisovan-related ancestry suggest that present-day Australo-Papuans most likely trace all of their AMH ancestry from the same migrant population as other contemporary non-Africans, though notably a small contribution (~2%) from a deeper AMH source cannot be entirely ruled out [30]. 

These genomic results are consistent with phylogeographic studies of mitogenomes and Y chromosomes [35,36,37,38], which show that lineages carried by modern Indigenous populations across Australia and New Guinea form a distinct genetic clade with respect to human populations from other continents, with initial separations between Australian and Papuan lineages occurring by around 50 kya [23]. Indigenous Austalo-Papuans carry a number of deeply divergent Y chromosome and mtDNA haplotypes that are distinct from those found in other human populations (e.g., mtDNA lineages M42, M27, M28, M29, Q, O, N13, S, P4, P5, P1; [20,21,22,23]). For example, the mtDNA macro-haplogroup R found in Aboriginal Australians contains distinct lineages (P5 and P4b, later classified as P11 clades by Nagle et al. (2017)) that diverged from their Papuan sister clades (P1, P2, and P4a, since classified as P4 by Nagle et al. (2017)) around 50 kya [22,23,24]. Similar divergence times are observed for a number of mtDNA [22,23,25,39] and Y chromosome lineages [27,40], and this striking phylogeographic partitioning has been interpreted as evidence for either a single AMH arrival into Sahul that rapidly fissioned into distinct New Guinea and Australian lineages [23], or instead for two separate historical arrivals into regions now situated in New Guinea and Australia (possibly via separate migration routes) [22,25]. 

To further investigate AMH migratory pathways into Sahul, Brucato and colleagues [41] compared 58 modern Papuan genomes to those from modern individuals from islands in northern (North Moluccas and Kei) and southern (Lesser Sundas and Flores) Wallacea, finding support for population models consistent with AMH arrival into Sahul via the Northern Route. A separate phylogeographic study by Purnomo and colleagues [39] also investigated this question by combining available mitogenomes from Indigenous Australians and Papuans with >300 mitogenomes sampled from 11 modern populations spread across Wallacea. Remarkably, nearly all Wallacean mtDNA lineages were nested within Papuan clades, with only a handful of deeply rooted Wallacean lineages forming outgroups to Australo-Papuan clades. Similar results were also observed amongst a large sample of Wallacean and Australo-Papuan Y chromosomes [40], and taken together, these results strongly suggest that modern Wallaceans derive much of their ancestry from mainland Papuans. Further investigation of Wallacean mtDNA clades pointed to two temporal TRMCA clusters at ~3 and ~15 kya, which could indicate possible timings for two separate influxes of Papuan migrants into Wallacea (see next section). Papuan mtDNA lineages also appear to have spread to neighbouring islands in Melanesia around these times [42], and a recent genome-wide study of the Philippines reported evidence of Papuan-related ancestry amongst populations in the south of this archipelago [43].

The findings from these recent phylogeographical studies strongly suggest that the movement of Papuan lineages may have reconfigured the genetic profile of AMH populations in Wallacea, with impacts likely extending into surrounding regions (Figure 2). Importantly, while the population models used in the genomic study of Brucato and colleagues accommodate Papuan back-migration into Wallacea [41]—suggesting that some of the AMH founder ancestries may have been preserved in modern populations—the possibility of Papuan backflow has generally been underappreciated in previous research to date and has important implications for our understanding of the initial AMH migrations (see last section).

## 3. Extensive Migrations and Mixing in Holocene Island Southeast Asia (ISEA)

The handful of prior population genomic studies that have investigated historical Wallacean demography have revealed a highly dynamic history, with multiple genetic ancestries being introduced into the region around the arrival of Austronesian-speaking seafarers during the late Holocene. One of the first genomic studies attempting to reconstruct the history of ISEA by Lipson and colleagues [38] used a variety of allele-frequency-based statistics (e.g., f-statistics and admixture graphs; [44]) to show that Indonesian populations from either side of Wallace’s Line contain distinct sources of genetic ancestry. Specifically, while all surveyed Indonesian populations contained an ancestry component related to modern Ami people from Taiwan, an additional component most closely related to modern-day Austroasiatic speakers from Mainland Southeast Asia (MSEA) was common amongst western Indonesians but absent from the easternmost Wallacean islands. The lack of evidence for Austroasiatic language and associated material culture across Indonesia led the authors to speculate that both Asian-related ancestries may have been introduced during the Holocene expansion of Austronesian-speaking seafarers that carried both ancestries through admixture events that predated their arrival. Further, Indonesian populations west of Wallace’s Line were also found to have a third ancestry component that was the shared contemporary ‘Negrito’ populations of the Philippines, but this ancestry was absent in all surveyed Wallacean populations (all from the Lesser Sundas), being effectively replaced instead by Papuan-related ancestry.

Highly consistent genetic patterns were also reported in a more recent study by Hudjashov and colleagues [45] which utilised an expanded set of ISEA populations that included groups from northern Wallacea. Analysis of haplotype sharing patterns revealed that west Indonesian populations (i.e., from islands west of Wallace’s Line) clustered with those from the Philippines (including Negrito groups), whereas Wallacean and Papuan populations formed a separate cluster. Each Wallacean population was found to share a significant proportion of their haplotypes with modern Papuans, as well as non-Negrito populations from the Philippines. The authors note that non-Negrito Filipino populations have previously been reported as providing suitable proxies for Austronesian migrants, such that this result may reflect the diffusion of the Austronesian language into local Philippine populations prior to migrants from the latter arriving and mixing with Wallaceans. Finally, while evidence for MSEA-related ancestry was inconsistent across Wallacean populations, admixture time estimates support the parallel introduction of MSEA- and Austronesian-related ancestry sources between ~1–2.5 kya. This suggests either the contemporaneous movements into Wallacea of two genetically distinct groups or a single origin admixed migrant group, around 1000 years after the earliest archaeological evidence for Austronesian arrival in the area.

These studies indicate that the genetic profile of modern Wallacea was largely determined by movements and mixing events in the late Holocene, possibly through the arrival of admixed Austronesian-speaking seafarers that carried multiple ancestry sources. Crucially, these interpretations implicitly assume that the Papuan-related ancestry observed in Wallaceans most likely reflects the AMH founder ancestry endemic to each island. However, the strict partitioning of Negrito and Papuan-related genomic ancestries reported for populations situated either west or east of Wallace’s Line, respectively, may instead signal the effective replacement of an ancient AMH genetic profile across Wallacea by incoming migrants bearing Papuan and potentially additional ‘Asian’ related ancestries. Indeed, it remains possible that the underlying mixing events may not have been local to each Wallacean island and could even have occurred outside of Wallacea. The extent to which these mixing events have decoupled Wallacean genetic profiles from their original island context remains an outstanding empirical question, and its resolution will be crucial for future genetic reconstructions of the deep human history of the region.

## 4. Wallacean Paleogenomic Studies

The complex population history of Wallacea evident in genetic studies suggests that ancient DNA (aDNA) research may provide the key to recovering key historical details. By situating genetic sequences within specific historical, geographical, and archaeological contexts, aDNA can illuminate aspects of human population history that may no longer be evident in modern genetic data. The field of ancient aDNA has developed rapidly since researchers first retrieved fragments of DNA from the desiccated muscle of an extinct subspecies of an Equus quagga museum specimen in 1984 [46]. In the past decade, continuing advancements in ancient DNA laboratory methods and sequencing technologies have facilitated the generation of genomic data on a population scale, launching modern paleogenomic research. While thousands of paleogenomes have now been generated from remains found in temperate climates, paleogenomes from humid tropical regions remain rare due to the comparatively poor preservation of genetic material [47]. Nonetheless, the first paleogenomic studies of individuals from Mainland Southeast Asia (MSEA) [48], Oceania [49,50,51,52], and ISEA [53] have begun to emerge in the past five years, including the first paleogenomes from Wallacea.

In 2021, the first Wallacean paleogenome was published from an ~8000-year-old female from Leang Panninge in southern Sulawesi [53], which notably predates the arrival of Austronesian seafarers to this region. Ancestry decomposition and *f*-statistic analyses showed that the Leang Panninge specimen has two distinct genetic ancestries, with one component that is genetically equidistant to both modern Indigenous Australians and Papuans, and another that is genetically closer to ancient East Asian and modern Andamanese (Onge) populations. Admixture graph models suggest that the Australo-Papuan-like component may reflect the genetic profile of the initial AMH migrants to ISEA, with subsequent mixing with an unknown Asian lineage resulting in the Leang Panninge specimen’s distinctive dual ancestry [53]. Interestingly, there was no evidence that this Neolithic genetic profile survived in any present-day populations from Sulawesi or elsewhere in Indonesia, pointing to its possible widespread replacement in Sulawesi (and potentially elsewhere in Wallacea) at some point during the interceding period.

A more recent study of 16 paleogenomes from multiple sites in Sulawesi, Moluccas, and Lesser Sundas, which range from several hundred to ~2300 years old, also found no evidence for genetic continuity between these 16 individuals and the Leang Panninge specimen [54], indicating that this genetic discontinuity was present within 1000 years of the initial Austronesian contact. All ancient individuals also had substantial Austronesian- and Papuan-related ancestries, and most also carried a distinct MSEA-related component, with ancient Moluccans being the exception. Notably, this MSEA component was present in contemporary Moluccan genomes, suggesting a staggered arrival of this ancestry across Wallacea. In contrast, admixture history analyses suggested that the MSEA- and Papuan-related ancestry components had mixed first for populations from the Lesser Sundas, prior to the introduction of Austronesian ancestry.

These paleogenomic studies provide the first direct support for the erosion of genetic profiles associated with founder AMH migrants across Wallacea through the introduction of external genetic ancestries. While the timing and ordering of the underlying mixing events appear to have varied across Wallacea, estimates of admixture times from ancient and modern genomes consistently point to initial mixing events occurring around 1–2.5 kya [38,45], with Moluccan paleogenomes indicating that MSEA ancestry was only introduced into the region after ~1 kya [54]. This suggests that the initial mixing events lagged behind the arrival of Austronesian seafarers by around 1000 years (based on the local archaeological evidence), though it is possible that admixture may have involved multiple events over a protracted time period across Wallacea, such that the estimates are no longer informative about these initial mixing events. Nonetheless, these results point to the arrival of Austronesians as potentially being the crucial period that transformed Wallacean ancestry, providing some hope that crucial details about the initial AMH settlement of this region might be retrievable if aDNA can be recovered from samples that precede Austronesian arrival. 

## 5. Archaic Hominin Introgression

The landmark sequencing of hominin genomes from two distant AMH relatives, namely Neanderthals and Denisovans, led to the momentous finding that these now-extinct hominin species had interbred with AMH as they dispersed from Africa across the rest of the planet [33,34,55,56]. Studies searching for hominin DNA sequences within contemporary human genomes have revealed that all present-day non-African human populations carry at least 2% Neanderthal genetic ancestry, with Denisovan DNA occurring at trace levels in South and East Asians [57,58] but being considerably more common (~2–6%) in populations living east of Wallace’s Line [32,34,59,60,61,62].

Ongoing investigations of the geographic patterning of hominin admixture in populations from ISEA and Sahul attest to a particularly complex set of historical interactions in this region, suggestive of multiple distinct mixing events occurring in situ across different islands [60,61,62,63]. Present-day Papuan populations carry DNA tracts from two genetically distinct hominin groups distantly related to the sequenced Denisovan individual, including one that may have occurred in Sahul [63], while contemporary populations from the Philippines carry more Denisovan-related ancestry than expected relative to their proportion of ‘Papuan’-related ancestry, suggesting an independent local mixing event that occurred after their separation from Australo-Papuans [33,61,64].

The high levels of Denisovan-related ancestry found across ISEA and Sahul is particularly surprising given that all currently identified Denisovan fossils come from mainland Asia [33]. However, the diverse regional fossil and stone tool records of ISEA suggest that at least three distinct hominin groups—namely, *H. erectus*, *H. floresiensis*, and *H. luzonensis*—were present in the region prior to the arrival of AMH in the region by 50–70 kya [1,2,3,4,5,6,65], but as yet, no DNA has been recovered from these specimens, meaning their genetic relationships to other hominins is currently unknown. This raises the intriguing possibility that some of the ISEA hominin specimens might actually represent a southern radiation of Denisovans, which had admixed with incoming AMH migrants [62]. To address this question, a recent study investigated if modern human populations in ISEA and New Guinea carried genetic admixture signals that were consistent with the local hominin fossil record. While signals consistent with Neanderthal- and Denisovan-related ancestry were widespread, no signals reconcilable with separate admixture events involving a genetically distinct hominin lineage were recovered, suggesting that the ISEA fossil record may indeed contain currently unidentified southern Denisovan relatives. Alternatively, the authors suggested that the Denisovan ancestry in human populations from ISEA and regions further east may instead come from a hominin group that still lacks fossil evidence or a sequenced genome, suggesting Sulawesi as a plausible location for further investigation.

## 6. Conclusions and Future Directions

Wallacea and Sahul are key regions of hominin evolution whose history remains only partially explored, partly due to conditions that have made fossils and archaeological records challenging to retrieve and preserve. In the past two decades, a small but growing number of genetic and phylogeographic studies from Wallacea and Sahul’s populations have started to explore the deep human history of this region, attempting to unravel the timing and location of the original human migrations and mixing events with now-extinct hominins. Notably, these studies have revealed a complex demographic history marked by multiple migrations that appear to have largely erased the deep genetic signals needed to properly resolve these fundamental questions. 

Moving forward, further genomic studies are sorely needed to determine the timing and origin of the migrations that brought Papuan-related ancestry into Wallacea and beyond, and to ascertain the extent to which modern Wallacean populations retain genetic information from the original AMH migrants. Notably, even if this genetic information has been retained, the complexity and extent of historical migrations impacting Wallacea may have sufficiently decoupled these genetic signals from their initial geographical context, to make the resolution of the migration routes impossible from modern genomes alone. Accordingly, the path ahead may ultimately require the generation of paleogenomes from AMH specimens across Wallacea and ISEA in the period preceding these mixing events. Similarly, determining the identity of hominin ancestry in modern populations east of Wallace’s Line may require ancient DNA or proteomic data from regional hominin fossils. Encouragingly, aDNA recovery from tropical regions has improved rapidly in recent times, suggesting that at least some of the paleogenomic data necessary to address these important questions may yet emerge in the foreseeable future.

## Figures and Tables

**Figure 1 genes-13-02373-f001:**
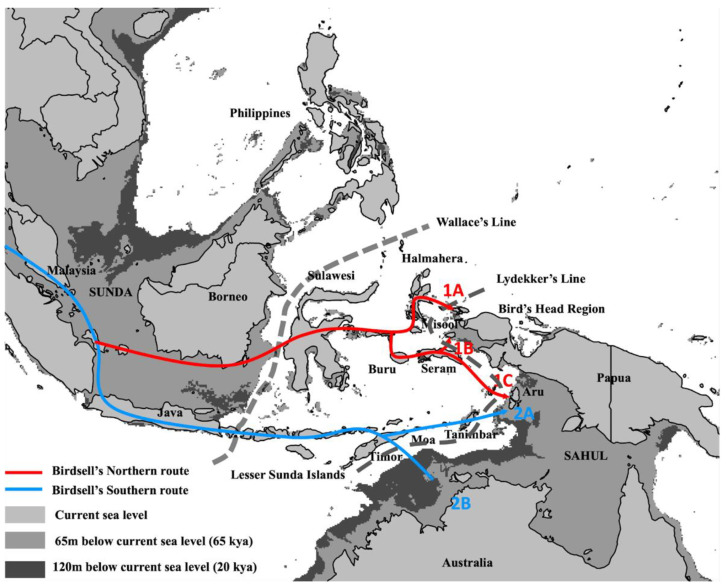
**Map of Wallacea and Sahul with palaeo-geographic reconstructions at lower sea levels** Wallacea lies between the former continents of Sundaland (to the west of Wallace’s Line) and Sahul (lying east of Lydekker’s Line), requiring incoming human migrants to undertake a series of interisland movements across the archipelago before arriving in Sahul (historical coastlines at multiple time periods are indicated by different grey shades). Two main routes through the Wallacea have been identified [19]. Migration along the Northern Route (red arrows) would have brought AMH from Borneo through Sulawesi, from which there were three potential entry points into Sahul: (1) through Halmahera before arriving at the Bird’s Head peninsula of West Papua (route 1A); (2) through Seram before arriving at Misool (route 1B); and (3) moving south into Aru (route 1C). Notably, both Misool and Aru were part of the Sahul continent based on sea level reconstructions at 65 kya (i.e., the oldest dated AMH-associated site in Australia [10]). In contrast, Southern Route migrants would have moved across the Lesser Sunda Islands (present-day Nusa Tenggara) and continued either north through Moa and Tanimbar before arriving on Aru (route 2A), or south directly to Australia (route 2B). Palaeo-geographical models comparing the two routes have favoured the Northern Route hypothesis, with route 1B having the most support based on analyses of island intervisibility [16] and a least-cost model [17]. Additionally, a study combining island intervisibility and oceanic drift modelling indicated that the Southern Route was also a viable option [12,18].

**Figure 2 genes-13-02373-f002:**
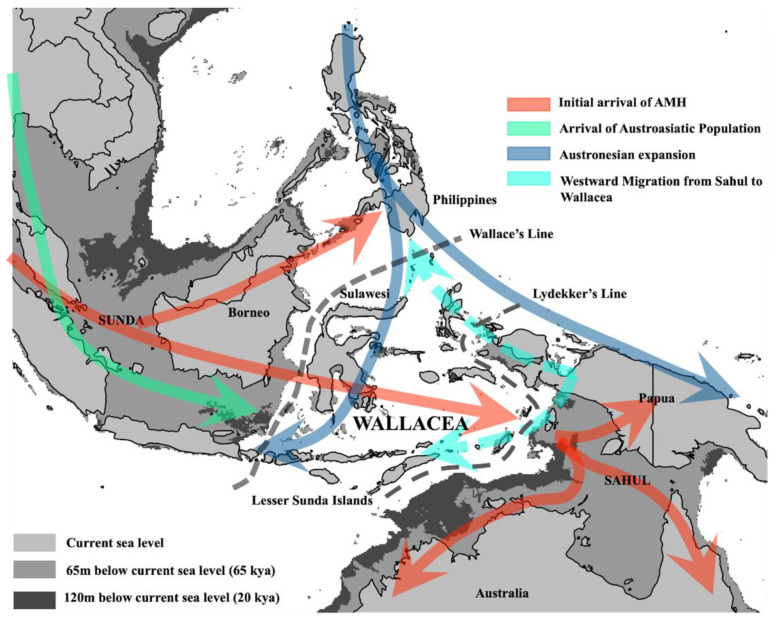
**Schematic representation of putative AMH population movements into ISEA and Sahul inferred from regional genetic data**. Genetic inferences suggest that the initial peopling of the region occurred around 50–60 kya, with the separation of Aboriginal Australian and New Guinea populations occurring around the same time (red arrows; noting that this split may have occurred prior to the settlement of Sahul). Later migrations associated with the arrival of Austronesian seafarers introduced Ami-related ancestry into ISEA (blue arrows), and possibly also introduced MSEA-related ancestries at the same time (though the latter may have come from a separate movement; green arrows). Recent studies have emphasised that the presence of Papuan-related ancestry across Wallacea and the southern Philippines likely represents the movement of Papuan lineages into these regions (light blue dashed arrows), possibly in two distinct movements around 3 kya (i.e., potentially tied with Austronesian arrival) and 15 kya [39,43]. Not shown is an additional, albeit relatively minor, South Asian ancestry component in western Indonesia [38,45], which appears to coincide with the establishment of trading between these regions around 2000 years ago.

## Data Availability

Not applicable.

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
