# Peer review of "Human Genetic Research in Wallacea and Sahul: Recent Findings and Future Prospects"

_genes, 2022, doi:10.3390/genes13122373_

Round 1

Reviewer 1 Report

The authors provide a review of the patterns of human migration within the Wallacea and Sahul geographic areas, and how genetic data has and has not completed this picture. I found the manuscript very well written. I have minor comments below that include a needed Intro paragraph to explain the goals/objectives/questions/gaps that the review will address, and a "future directions" section that provides specific ideas on future directions. Everything in between fits a standard "review" presentation.

Lines 69-73:  I’m struggling to figure out how this paragraph fits in. The preceding paragraph reviews evidence of migration patterns from non-genetic work, and the following sections talk about genetic studies, of which we have multiple datasets. Thus, this paragraph suggesting that genetic analyses “could provide an alternative tool” implies that this is something that should/could be done…even though it already has and will be reviewed forthcoming. It’s confusing. 

The overall objective, goals, questions, and innovation is missing here. I would develop this paragraph a bit more to end the Intro like perspective/review papers do. For example, maybe the authors are trying to use this paragraph to set up what the BIG questions and hypotheses are and what genetic studies would do to address these questions, and thus after setting up these specific ideas and questions, they have the typical  “Here, we conduct a review of the literature in presenting genetic analyses to date to determine how well they address these questions, and what gaps still remain” (or something like that?). Otherwise, it’s honestly not very clear what the objective of the manuscript is…review? Primary data? And thus, this last paragraph of the Intro needs to do that. I got about 4 pages in before I realized this was a review with no new data/analyses. 

Line 198: I would suggest changing the phrase “but their progeny went extinct” to “but derived lineages from these introgression events went extinct”. Lineages and species can go extinct, but it is awkward to refer to one’s “progeny” going extinct.

Line 211: I would suggest changing the phrase “after the success of the quagga sequence” (which does not make sense) to “after sequencing the quagga genome”. 

Line 235: I would suggest changing the phrase “it is far from impossible” (which is not quantifiable) to “it is possible”.

Line 261: “Austronesian” instead of “Austonesian”?

“Future directions” section: This section reads more like a recap. The last sentence provides the point that we need to “do more”, but that’s not really a “future direction”. This section needs to feed back to the Intro to tell us what needs to be done next. The authors could be innovative here in telling the audience: what are the outstanding specific questions? What groups and what geographic regions are needed to fill in the gaps genetically? What could be some interesting finds?…speculate a bit to catch the audience interest at the end!

Figures:

Figure 1: It’s a bit difficult to see the colored lines and text in Figure 1, specifically the “1B”, “1C”, “2A”, and “2B” are lost in the background. The legend bar is also not intuitive and not explained in the caption. 65 kya and 20 kya are clear, but what are “-65 m” and “-120m”, respectively?

Figure 2: As is true with this and others, the authors are providing a lot of text in the figure caption that is not describing the figure itself, but instead “information content”. This figure is a perfect example of that, where it is simply a figure of Wallacea/Sahul with shaded areas of arrival/migration. That’s all that needs to be said for a caption, the rest is starting to explain and detail where the info comes from and interpretation of data, which really should be in the main text only. Related, this figure did not reproduce well. It is impossible to see the shades of grey in the legend and on the map, when there are also other shades of grey outlining the geographic regions too. The figure needs to involve color and other better visuals to show these different contrasts laid on top of each other.

Reviewer 2 Report

This paper provides an overview of genetic variation in Southeast Asia and Sahul and its contribution towards reconstructing the demographic history of the region, both of archaic and modern humans. It is a clear, well-balanced review that summarises the main areas of debate and the main contributions to solving the questions from genetics and genomics, drawing on genomic data, uniparental data and the little ancient DNA there is. I believe it will be very useful to researchers. There are just a few minor omissions and some corrections needed with details and figures.

There are major labelling problems with both figures. The Philippines is spelled incorrectly, Wallace’s Line is called ‘Wallacea’s Line’ (and is not actually given on the maps), Lydekker’s Line is called ‘Lydekker’s lines’, there is a spurious point after Halmahera, and Present is rendered ‘Pres.ent’. Fantone Bank is mentioned in the legend to Figure 1 but not labelled. The Birdsell citation is missing from the legend to Figure 1. Moreover, all of the detail of arrows etc. is completely missing from Figure 2, at least in my copy.

‘Northern Route’ should be capitalised consistently throughout.

Reference 25 seems inappropriate since it has no reference to the genetic data just mentioned and is 20 years old. It would be more suitable to cite a paper that describes the genetic data, but they do not actually support the statement about a mainland origin for haplogroups B and E – in fact mtDNA haplogroup M7c appears to be the main Austronesian lineage with a mainland source – so this should be qualified. See Soares et al. Resolving the ancestry of Austronesian‑speaking populations Hum Genet (2016) DOI 10.1007/s00439-015-1620-z (for mtDNA hg M7c, but also summarising B and E) – the latter would make more suitable citation here.

p.3, line 99: It is a true that a single individual with P1 has been identified in the Philippines, but amongst the Maranao, not the ‘negrito’ groups – the latter carry the main Philippine haplogroup P lineages, which belong to subgroups P9 and P10 and do not occur in Sahul. (Also, in passing, the Philippines are not part of Wallacea, as implied here.)

There is a significant omission of Y-chromosome evidence in dividing the sections into mtDNA and genome-wide data – in particular, Bergström et al. Deep roots for Aboriginal Australian Y chromosomes. Curr Biol (2016) doi.org/10.1016/j.cub.2016.01.028 should surely be discussed and cited.

p.4, line 115: It is incorrect to cite Mallick et al. (2016) as supporting an earlier out of Africa dispersal contributing to Australians – they explicitly argue against it. The authors may be thinking of Pagani et al. (2016) which was published side by side the Mallick paper and did make the case, but for New Guineans rather than Australians.

p.5: ‘hunter gatherer-like’ might be better as hunter–gatherer-like.

p.6, line 208: ‘Equus quagga’ is technically correct, but loses the sense that it was the extinct subspecies Equus quagga quagga that led to the breakthrough.

On p.7, there are several cases or spurious capitalisations (Hunter-gatherer, Individual) an Austronesian is misspelled ‘Austroneasian’. It also seems unclear what the place of the ‘equally’ is on line 241, and ‘interceding’ should be intervening. Also, the use of ‘unravelled’ implies the opposite of what the authors seem to intend (and the way it is used in the next paragraph) – perhaps ‘unresolved’ would be better.
